# Multi-view Self-Supervised Contrastive Learning for Multivariate Time Series

## ABSTRACT

Learning semantic-rich representations from unlabeled time series data with intricate dynamics is a notable challenge. Traditional contrastive learning techniques predominantly focus on segment-level augmentations through time slicing, a practice that, while valuable, often results in sampling bias and suboptimal performance due to the loss of global context. Furthermore, they typically disregard the vital frequency information to enrich data representations. To this end, we propose a novel self-supervised general-purpose framework called **T**emporal-**F**requency and **C**ontextual **C**onsistency (**TFCC**). Specifically, This framework first performs two instance-level augmentation families over the entire series to capture nuanced representations alongside critical long-term dependencies. Then, TFCC advances by initiating dual cross-view forecasting tasks between the original series and its augmented counterpart in both time and frequency dimensions to learn robust representations. Finally, three specially designed consistency modules —temporal, frequency, and temporal-frequency— aid in further developing discriminative representations on top of the learned robust representations. Extensive experiments on multiple benchmark datasets demonstrate TFCC's superiority over the state-of-the-art classification and forecasting methods and exhibit exceptional efficiency in semi-supervised and transfer learning scenarios. Code, data, and model checkpoints will be released after the review period.

## CCS CONCEPTS

• **Computing methodologies** → **Unsupervised learning**; *Learning latent representation.*

## KEYWORDS

Time series analysis, Time-frequency mining, Multi-view self-supervised learning, Contrastive learning

## 1 INTRODUCTION

Time series analysis plays a vital role in real-world scenarios such as healthcare, financial markets, and energy. These data are typically collected by Internet of Things (IoT) sensors and capture the dynamics of variables over time. However, acquiring such label data often entails significant time and financial investment. The label sparsity hampers traditional supervised learning approaches owing to insufficient training data and impacts the downstream

tasks [22]. The endeavor to derive generalized representations from such constrained time series data presents an intricate challenge; thus, unsupervised representation learning in time series analysis paves the way for label-scarce scenarios. Lately, numerous studies [5, 9, 10, 26] have utilized contrastive loss to uncover the intrinsic patterns of time series, but they still face some notable limitations.

First, **segment-level representations introduce sampling bias**. Previous efforts, such as TNC [18], CPC [15], TS-TCC [5], and CA-TCC [6], have yielded charming contributions, driving significant strides in representation learning. These methods mainly apply segment-level sampling policy and create contrastive pairs along the temporal axis, which may fall short of capturing the global context and undermine the preservation of temporal dependencies within time series [25]. Segment-level methods are highly dependent on the way to construct contrastive pairs, which usually assume neighboring segments as positive pairs and distant ones as negative, may not perform well in long-term scenarios and may not be able to capture the complete semantic information, thus affecting the stability and effectiveness of downstream tasks [22].

Next, **frequency domain information is not fully explored**. The aforementioned approaches have exhibited their strong ability for self-supervised representation learning, but they all overlook the frequency properties and the temporal-spectral affinities in time series, limiting the discriminative and expressive representation learning. CRT [25], CoST [20], and BTSF [22] provided valuable insights for mining frequency information, but only implemented augmentations in the time domain. The time domain focuses on sequential order and continuity, whereas the frequency domain emphasizes periodic patterns. Solely focusing on time-domain information often neglects important features, which can significantly hinder model performance, particularly in complex and noisy series. TF-C [26] considered the time and frequency augmentations, yet it lacks considerations for phase augmentation and sequence context information, indicating it was not fully mining frequency patterns and ill-suited for prediction tasks. Exploring frequency information can unearth vital insights and improve the effectiveness and generalizability of learned representations. However, how can we fully capture frequency domain information? Extracting robust information through prediction has been proven effective in the time domain [6], but what about the frequency domain?

To address these challenges, we present TFCC, a novel multi-view self-supervised contrastive learning framework that enhances time series representation learning through three aspects. First, we applied instance-level augmentations to create contrastive pairs in both time and frequency domains, especially perturbating the frequency spectrum, which would not alter the raw properties of the entire series and effectively reduce false negatives and false positives. Additionally, we designed two cross-view forecasting modules in the time and frequency domains to encourage the model to learn robust representation using the past augmented samples to predict the future of the original samples without data loss and

vice versa, which may differ from TS-TCC. To our knowledge, this is the first work that explores phase perturbation augmentation and performs prediction to capture spectrum patterns in the frequency domain. Last, we leverage three consistency modules in latent space: two contextual consistencies and one temporal-frequency consistency. Contextual consistency aims to maximize intra-sequence similarity and minimize inter-sequence similarity in the same domain, fostering discriminative representation learning. Temporal-frequency consistency is designed to pull together the time-based and frequency-based representations of the same series and push away the representations of different series in the shared latent space, achieving mutual learning from cross-domain interactions. As a result, TFCC utilizes multiple views to empower the exploration and extraction of generalizable time series representations and capture the embeddings invariant in various temporal dynamics and semantic meaning, leading to superior performance in downstream tasks, and serving as a vehicle for semi-supervised classification domain adaption and label-scarce scenarios.

In summary, the contributions of our work are as follows.

- A novel multi-view contrastive learning framework is proposed for unsupervised general-purpose representation learning in time series data.
- Two novel instance-level augmentation families in the time and frequency domains especially the phase perturbations, are first proposed to preserve global context and capture long-term dependencies.
- TFCC aims to learn universal representations. Two novel cross-view forecasting tasks are designed to learn robust representations. To our knowledge, we are the first to conduct predictions to learn robust representations in the frequency domain. TFCC promotes discriminative learning with two contextual consistencies and a temporal-frequency consistency that aligns time-based and frequency-based embeddings through cross-domain interactions in the latent space.
- Extensive experiments demonstrate that TFCC surpasses existing state-of-the-art methods in classification and forecasting tasks, and prove its effectiveness in semi-supervised learning and transfer learning settings.

## 2 RELATED WORK

**Self-supervised Learning for Time Series**. Self-supervised representation learning for time series has gained significant growth, but there remains considerable scope for improvement in this area. Inspired by the well-established self-supervised learning methods in computer vision and natural language processing, recent studies primarily utilize contrastive learning frameworks for time series representation learning. For instance, TST [24] adopted masked signal prediction as an auxiliary task for transformer model pre-training. TS-TCC [5] constructed a cross-view prediction on weak and strong augmentations in the temporal module to learn robust representations and proposes a context consistency module to learn discriminative representations. TNC [18] maintains temporal consistency by distinguishing between neighboring and non-neighboring signals. TS2Vec [23] optimizes hierarchical contrast learning at multiple scales. However, these time-based methods are often inadequate for long time series, failing to capture long-term

dependencies effectively [22]. Besides, the effectiveness will significantly diminish when applied to downstream tasks involving periodic time series [25].

**Temporal and Frequency Contrasting**. Exploring the frequency domain can yield extra insights, and few studies have considered transformation invariance and investigated this area, but they may still have some aspects that could be improved. CoST [20] proposed contrastive learning in time and frequency domains to learn disentangled seasonal-trend representations for time series forecasting. BTSF [22] and CRT [25] also included the frequency domain, but they only perform augmentations in the time domain. Similarly, TF-C [26] and TS-TFC [11] incorporate frequency features. Nevertheless, TF-C failed to consider phase perturbation and contextual invariance, while TS-TFC overlooks the use of augmentations, which may affect their model's generalization. In this work, we first present two instance-level augmentation families to enrich the pre-trained models, and two cross-view forecasting tasks to learn robust representations, especially by introducing spectral perturbations and prediction strategy in the frequency domain. Besides, we propose contextual consistency and temporal-frequency consistency modules to capture discriminative information by cross-domain correlations.

## 3 METHOD

Our proposed TFCC framework is shown in Figure 1.

### 3.1 Time Series Data Augmentation

Data augmentation is vital in contrastive learning for enhancing data diversity [3, 5, 12]. Contrastive learning aims to maximize the similarity between different views of the same sample while minimizing similarity with other different samples, making similar instances grouped and dissimilar instances separated. We create positive pairs by matching the original series with its augmented counterpart, treating other samples as negative pairs. Wen [19] thoroughly reviewed time series augmentation techniques. Data augmentation usually introduces synthetic disturbances that reflect the complexity of time series, which mitigates overfitting risks and improves model robustness and generalization. However, current methods often lose global semantic information and fail to capture long-term dependencies due to segment-level sampling bias.[22].

This work presents a novel data augmentation strategy designed to maintain data's raw properties and capture global context. We leverage various invariant transformations to build a time-based augmentation family (TAF) and a frequency-based augmentation family (FAF), enriching the representation learning during pre-training. For TAF, we utilize instance-level temporal augmentations, including scaling, jittering, and dropout mask to generate diverse data while keeping the entire series integrity, all well-recognized in [17, 22, 26]. Given a sample $x$, we input the sequence $x^T$ into TAF to produce positive samples $x_{pos}^T$. Negative samples $x_{neg}^T$ are generated by randomly selecting other variables. For the FAF, we introduce perturbations to the spectrum by adding or removing frequency components and phase and amplitude perturbations. $x^F$ is derived through the Fast Fourier Transformation (FFT) [14] and then create $x_{pos}^F$ by applying FAF. Negative samples $x_{neg}^F$ are obtained from other variables' frequency. In experiments, the model will randomly

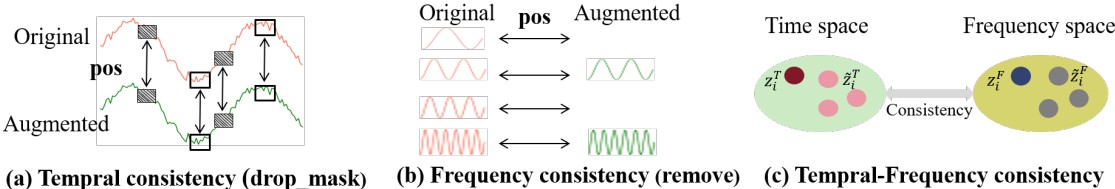

**Figure 1: The diagram of our universal self-supervised representation learning framework TFCC for multivariate time series.**

**(a) Tempral consistency (drop_mask)**     **(b) Frequency consistency (remove)**     **(c) Tempral-Frequency consistency**

**Figure 2: Positive pair**

select one time-based augmentation from TAF and one frequency-based augmentation from FAF to increase data diversity to enhance the model's ability to learn time series representations.

## 3.2 Cross-view prediction

**Temporal domain** Inspired by [5], we developed our temporal contrasting module to capture the latent temporal features. For a given input sample $x_i$, $i$ is the timestamp of a time series sample, we generate an augmentation set $X_i^T$ using TAF: $x_i^T \to X_i^T$. Each member $\tilde{x}_i^T \in X_i^T$ is augmented from $x_i^T$. We randomly select a member $\tilde{x}_i^T$ to generate a positive pair with the original sample, and expose the model to complex, missing, and unstable temporal dynamics to derive more robust embeddings. The original and the augmented views are all passed through an encoder consisting of three convolutional layers to obtain the latent embeddings: $h_i^T = \text{Encoder}\left(x_i^T\right)$, $\tilde{h}_i^T = \text{Encoder}\left(\tilde{x}_i^T\right)$. Transformer as an autoregressive model is

utilized to yield contextual vectors $z_t^T = \text{Transformer}\left(h_{0 \le i \le t}^T\right)$, $\tilde{z}_t^T = \text{Transformer}\left(\tilde{h}_{0 \le i \le t}^T\right)$.

We introduce a challenging cross-view prediction task employing a bilinear prediction model that utilizes the context of $z_t^T$ to predict the future M timestamp-augmented sample $\tilde{h}_{t+m}^T$ ($1 \le m \le M$), and use $\tilde{z}_t^T$ to predict $h_{t+m}^T$, which means the mutual information learning between the input $x_{t+m}$ and $z_t$, $f\left(h_{t+m}, z_t\right) = \exp\left(\left(\mathcal{G}_m\left(z_t\right)\right)^T h_{t+m}\right)$, where $\mathcal{G}_m$ is a linear function that maps $z_t$ back into the same dimension as $h_t$. Two contrastive losses $L^T$ and $\tilde{L}^T$ are designed to maximize the dot product between the original and the predicted of the same sample while minimizing the dot product with other samples $\mathcal{N}_{t,m}$ in the batch, computed as:

$$L^T = -\frac{1}{M}\sum_{m=1}^{M}\log\frac{\exp\left(\left(\mathcal{G}_m(z_t^T)\right)^T h_{t+m}^T\right)}{\sum_{n \in \mathcal{N}_{t,m}}\exp\left(\left(\mathcal{G}_m(z_t^T)\right)^T h_n^T\right)} \quad (1)$$

$$\tilde{L}^T = -\frac{1}{M} \sum_{m=1}^{M} \log \frac{\exp\left((\mathcal{G}_m(\tilde{z}_t^T))^T h_{t+m}^T\right)}{\sum_{n \in N_{t,m}} \exp\left((\mathcal{G}_m(\tilde{z}_t^T))^T h_n^T\right)} \quad (2)$$

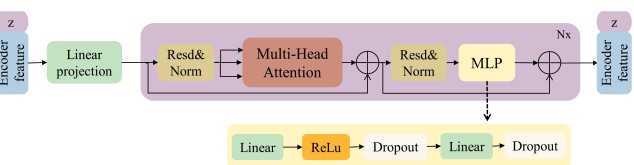

**Figure 3: Architecture of the Transformer in our framework. The token $z$ in the output is sent to the non-linear *Projection* used in temporal and frequency contrasting modules.**

Figure 3 showcases the architecture of the Transformer used in our work, which is the same as that in [5]. It mainly consists of consecutive Multi-Head Attention (MHA) blocks and an MLP block. The MLP consists of two fully connected layers with a nonlinear ReLU function and filter, as well as two Dropout in the middle. Our Transformer employs pre-normalized residual connectivity, which produces more stable gradients. We add a token $z$ into the module and then output its state as a representation context. The Transformer operation begins by taking the features $h_{\leq t+m}$ and passing them through a linear projection layer, which transforms these features into a hidden dimension, represented as $Projection : \mathbb{R}^{d \to k}$. The result of this linear projection is then fed into the Transformer, resulting in $Projection, \quad h' \in \mathbb{R}^k$. Subsequently, we combine the context vector with the feature vector $h'$, resulting in the input features becoming $\phi_0 = [z; h']$, where the subscript 0 signifies that it serves as the input to the first layer. Following this, we propagate $\phi_0$ through the Transformer layers using the equations below:

$$\phi_\ell' = \text{MH}\Lambda\left(\text{Norm}\left(\phi_{\ell-1}\right)\right) + \phi_{\ell-1}, \quad 1 \leq \ell \leq L \quad (3)$$

$$\phi_\ell = \text{MLP}\left(\text{Norm}\left(\phi_\ell'\right)\right) + \phi_\ell', \quad 1 \leq \ell \leq L \quad (4)$$

**Frequency domain** Time and frequency domains are two different views of the same sequence, and the latter can offer another perspective for uncovering spectral characteristics in time series [25]. Recent studies like CoST, BTSF, TFDNet, and TS-TFC [11] have examined frequency but missed out on frequency augmentations. Our work is one of the few to explore frequency-based augmentations and the first to perturb phase and amplitude. When perturbing the phase of $x$, we first extract its original phase, then generate a random perturbation controlled by perturb_ratio, and sum them up to get the final perturbed spectrum. For sample $x_i$, the frequency spectrum $x_i^F$ is generated from $x_i^T$ through FFT. For $x_i^F$, an augmentation set $X_i^F$ is built using FAF: $x_i^F \to X_i^F$. We utilize the same encoder and Transformer to map $x_i^F$ to a frequency-based embeddings $z_i^F = Transformer(Encoder(x_i^F))$, $\tilde{z}_i^F = Transformer(Encoder(\tilde{x}_i^F))$. We are the first to design cross-view prediction to capture frequency patterns in the frequency domain, the process of which is similar to that in the time domain, $L^F$ and $\tilde{L}^F$ are computed by:

$$L^F = -\frac{1}{M} \sum_{m=1}^{M} \log \frac{\exp\left((\mathcal{G}_m(z_t^F))^F h_{t+m}^F\right)}{\sum_{n \in N_{t,m}} \exp\left((\mathcal{G}_m(z_t^F))^F h_n^F\right)} \quad (5)$$

$$\tilde{L}^F = -\frac{1}{M} \sum_{m=1}^{M} \log \frac{\exp\left((\mathcal{G}_m(\tilde{z}_t^F))^F h_{t+m}^F\right)}{\sum_{n \in N_{t,m}} \exp\left((\mathcal{G}_m(\tilde{z}_t^F))^F h_n^F\right)} \quad (6)$$

## 3.3 Contextual Consistency

Prior studies have delved into contextual consistency [4, 5] but lack exploration of frequency consistency, which is complemented by our work.

**Temporal Consistency**. A batch consisting of N input samples, each has two contexts — one is from the original and the other is from the augmented, and thus get 2N contexts $\left[c_i^T, \tilde{c}_i^T, c_j^T, \tilde{c}_j^T, \cdots, c_N^T, \tilde{c}_N^T\right]$. For sample $x_i$, $c_i^T$ and $\tilde{c}_i^T$ are nonlinearly transformed by $c_i^T = Projection(z_t^T)$, $\tilde{c}_i^T = Projection(\tilde{z}_t^T)$. $c_i^T$ and $\tilde{c}_i^T$ are treated as a positive pair (Figure 2 (a)), while the remaining 2(N-1) contexts within the batch are categorized as negative pairs, collectively forming $c^{neg} : c^- \in c^{neg}$. As such, in this context, we employ $\mathcal{D}$ as the distance function to maximize the similarity between positive pairs while minimizing the similarity between negative pairs, enabling the representations to be discriminative. The temporal contextual consistency loss is defined as:

$$L_C^T = \mathcal{D}(c_i^T, \tilde{c}_i^T) = -\sum_{i=1}^{N} \log \frac{\exp\left(\text{sim}\left(c_i^T, \tilde{c}_i^T\right)/\tau\right)}{\exp\left(\text{sim}\left(c_i^T, \tilde{c}_i^T\right)/\tau\right) + \sum_{c^- \in c^{neg}} \exp\left(\text{sim}\left(c_i^T, c^-\right)/\tau\right)} \quad (7)$$

where $\text{sim}(\boldsymbol{u}, \boldsymbol{v}) = \boldsymbol{u}^T \boldsymbol{v}/\|\boldsymbol{u}\|\|\boldsymbol{v}\|$ is the cosine similarity, $\tau$ is the degree of pull and push as the temperature parameter and shares the same value throughout all experiments.

**Frequency Consistency**. We implement frequency consistency between the original and its augmented views to learn underlying patterns that the time domain might miss. We assume the model can learn similar features from the original spectrum $x_i^F$ and a perturbed spectrum $\tilde{x}_i^F$, thus the embeddings $c_i^F = Projection(z_i^F)$ and $\tilde{c}_i^F = Projection(\tilde{z}_i^F)$, i.e., $(c_i^F, \tilde{c}_i^F)$ can be denoted as a positive pair (Figure 2 (b)), similar to temporal consistency, $c^{neg} : c^- \in c^{neg}$, total 2(N-1) within the same batch. The frequency contextual consistency loss can be calculated as:

$$L_C^F = \mathcal{D}(c_i^T, \tilde{c}_i^T) = -\sum_{i=1}^{N} \log \frac{\exp\left(\text{sim}\left(c_i^F, \tilde{c}_i^F\right)/\tau\right)}{\exp\left(\text{sim}\left(c_i^F, \tilde{c}_i^F\right)/\tau\right) + \sum_{c^- \in c^{neg}} \exp\left(\text{sim}\left(c_i^F, c^-\right)/\tau\right)} \quad (8)$$

## 3.4 Temporal-Frequency Consistency

In the latent space, time and frequency embeddings from the same sample intuitively should be closer than other different samples. Inspired by this, we propose the temporal and frequency consistency, shown in Figure 2(c). For each sample $x_i$, we generate four embeddings: time-based original $c_i^T$, frequency-based original $c_i^F$, and their augmented version $\tilde{c}_i^T$ and $\tilde{c}_i^F$. These embeddings, derived from the data's temporal and frequency characteristics, emphasize both aspects' importance in time series analysis. Hence, we input both time and frequency domains into the model to strengthen the

cross-domain interactions and enhance the learning of temporal and spectral details.

To maintain consistency in time-frequency space, we introduce a consistency loss, $L_C^{TF}$, to measure the distance between time and frequency embeddings. We utilize $\mathcal{D}_i^{TF} = \mathcal{D}(c_i^T, c_i^F)$ to represent the distance between $c_i^T$ and $c_i^F$. Similarly, we define $\mathcal{D}_i^{T\tilde{F}}$, $\mathcal{D}_i^{\tilde{T}F}$, $\mathcal{D}_i^{\tilde{T}\tilde{F}}$. Notably, we only focus on cross-domain distances and exclude within-domain distances (i.e., the distance between $c_i^T$ and $\tilde{c}_i^T$, the distance between $c_i^F$ and $\tilde{c}_i^F$), which we have already calculated in $L_C^T$ and $L_C^F$.

Then, from $\mathcal{D}_i^{TF}$ and $\mathcal{D}_i^{\tilde{T}F}$, we can observe they involve $c_i^T$, $c_i^F$, $c_i^{\tilde{T}}$ three embeddings. $c_i^T$ and $c_i^F$ are from the same original sample $x_i$, while $c_i^{\tilde{T}}$ is from the time-augmented $\tilde{x}_i^T$, thus $c_i^F$ should be closer to $c_i^T$ than $c_i^{\tilde{T}}$. Therefore, we prompt the model to learn smaller $\mathcal{D}_i^{TF}$ than $\mathcal{D}_i^{\tilde{T}F}$. Similarly, $\mathcal{D}_i^{TF}$ should also be smaller than $\mathcal{D}_i^{T\tilde{F}}$ and $\mathcal{D}_i^{\tilde{T}\tilde{F}}$. Triplet loss widely used in [8, 16] is designed as the temporal-frequency consistency loss, enabling the model to learn smaller $\mathcal{D}_i^{TF}$. We calculate the consistency loss $L_C^{TF}$ by:

$$L_C^{TF} = \sum_{i=1}^{N} \left( \sum_{\mathcal{D}^{pair}} (\varepsilon + \mathcal{D}_i^{TF} - \mathcal{D}_i^{pair}) \right), \mathcal{D}^{pair} \in (\mathcal{D}_i^{\tilde{T}F}, \mathcal{D}_i^{T\tilde{F}}, \mathcal{D}_i^{\tilde{T}\tilde{F}}) \quad (9)$$

where $\varepsilon$ is a constant and set to 1. $\mathcal{D}^{pair}$ denotes the distance between the time embeddings $c_i^T$, $c_i^{\tilde{T}}$ and the frequency embeddings $c_i^F$, $c^{\tilde{F}}$. Each pair has at least one embedding from the augmented views. The $L_C^{TF}$ loss motivates the pre-training model to learn the consistency between the time and frequency embeddings during model optimization.

## 3.5 Overall Loss Function

Our self-supervised pre-training process has a total of seven losses. First, $L^T$, $\tilde{L}^T$, $L^F$, $\tilde{L}^F$ are introduced for the cross-view prediction to acquire robust representations. Second, $L_C^T$ and $L_C^F$ are used to capture embedded information that is invariant to augmentations and extract discriminative representations. Last, the time-frequency consistency loss, $L_C^{TF}$, ensures consistency between time and frequency latent embeddings. In summary, the overall loss is:

$$L_{TFCC} = \lambda 1 \cdot (L^T + \tilde{L}^T + L^F + \tilde{L}^F) + \lambda 2 \cdot (L_C^T + L_C^F) + \lambda 3 \cdot L_C^{TF} \quad (10)$$

where $\lambda 1$, $\lambda 2$, and $\lambda 3$ are fixed hyperparameters indicating the relative weight of each loss.

## 4 EXPERIMENTS

### 4.1 Experimental Setup

We conducted extensive experiments to assess TFCC's performance and performed ablation studies to highlight each component's contribution to the outcome. More detailed descriptions of augmentation methods and experiments are in Supplementary Materials.

**Datasets**. Classification (1) **Epilepsy** was conducted preprocessing following previous works[5, 22]. The original dataset featured 5 classes, but 4 did not contain epileptic seizures. Therefore, we

merged them into a single class and treated it as a binary classification problem. (2) **HAR** contains six human activities: walking, walking upstairs, walking downstairs, standing, sitting, and lying down collected by smartphones [1]. They use a waist-mounted Samsung Galaxy S2 device to record the sustained activity of each subject, with a 50Hz sampling rate. (3) **Sleep-EDF** records multi-sleep modes, utilized a single EEG channel, sampled at a rate of 100 Hz[7], which is designed to categorize the input EEG signals into one of the following categories: wake (W), non-rapid eye movement (N1, N2, N3), and rapid eye movement.

Forecasting (1) **ETT**[1] comprises two distinct granularities: two hourly-level datasets (ETTh) and one 15-minute-level dataset (ETTm). The data was collected from separated countries in China for long-sequence time series forecasting. Each dataset contains target values of 6 load features and "oil temperature" from July 2016 to July 2018. (2) **Weather**[2] contains 11 local climatological features focusing on 1600 locations in the U.S. over 4 years from 2010 to 2013.

**Implementation Details**. We run all experiments on one GeForce RTX 3090 Ti GPU, running 64-bit Linux 5.15.0-56-generic, and perform a 60/20/20 train/validation/test split. We set the epoch size to 50, the batch size to 128, Adam optimizer, a learning rate of 3e-4, weight decay of 1e-4, and dropout to 0.2, $\beta_1 = 0.5$, $\beta_2 = 0.55$. For temporal augmentations, the drop_mask ratio is 0.2, while the scale ratio and jitter ratio are different according to different datasets($scale_{Ep}$=0.001, $jitter_{Ep}$=0.001, $scale_{EDF}$=1.5, $jitter_{EDF}$=2, $scale_{HAR}$=1.1, $jitter_{HAR}$=0.8). For frequency augmentations, the pertub_ratio is 0.1. The number of heads in the Transformer is 8.

### 4.2 Comparison with Baseline Approaches

**Baselines**. Classification: Random Init.: A linear classifier on a randomly initialized encoder. Supervised: Supervised training of both encoder and classifier. CPC [15]. SimCLR [4]. MHCCL [13]. TS-TCC [5]. TSTCC-Floss [21]. TF-C [26]. BTSF [22]. CA-TCC [6].

Forecasting: 1) Representation learning: TNC[18]. TS-TCC. TS2Vec [23]. CoST[20]. BTSF [22]. 2) End-to-end models: TCN [2]. Informer [27].

**Results**. Classification: We evaluate the performance of these methods on three widely used datasets. For a fair comparison, we adhere to previous linear benchmark evaluation models [4, 6] and measure how well the learned representations are used to classify hidden states with two metrics: accuracy=ACC = $\frac{TP+TN}{TP+TN+FP+FN}$, and the macro-averaged F1-score MF1= $\frac{2 \times PR}{P+R}$. Before training the linear classifier, we froze the self-supervised pre-trained encoder. Table 1 summarized the results, with the best results in **bold** and the second-best underlined. Our TFCC outperforms all baselines in all datasets, including the supervised method. This superior performance is ascribed to the instance-level augmentations and multi-view contrasting strategy, helping to capture the long-term dependencies and the learned transformation-invariant representations. Additionally, our TFCC surpasses three TS-TCC-based methods (TS-TCC, TSTCC-Floss, CA-TCC), especially in the HAR dataset, with improvements of 2.36%, 1.87%, and 1.05%, and outperforms the TF-C and BTSF by 4.58% and 7.29%, which illustrates TFCC's effectiveness and superiority.

---

[1]https://github.com/zhouhaoyi/ETDataset
[2]https://www.ncei.noaa.gov/data/local-climatological-data/

**Table 1: Classification results. Best results are highlighted in bold, while the second-best is underlined**

| Methods | Epilepsy | | HAR | | Sleep-EDF | |
|---|---|---|---|---|---|---|
| | Accuracy | MF1 | Accuracy | MF1 | Accuracy | MF1 |
| Random Init. | 90.26±1.77 | 81.12±4.22 | 57.89±5.13 | 55.45±5.49 | 35.61±6.96 | 23.80±7.96 |
| Supervised | 96.66±0.24 | 94.52±0.43 | 90.14±2.49 | 90.31±2.24 | 83.41±1.44 | 74.78±0.86 |
| CPC | 96.61±0.43 | 94.44±0.69 | 83.85±1.51 | 83.27±1.66 | 82.82±1.68 | 73.94±1.75 |
| SimCLR | 96.05±0.34 | 93.53±0.63 | 80.97±2.46 | 80.19±2.64 | 78.91±3.11 | 68.60±2.71 |
| MHCCL | 97.85±0.49 | 95.44±0.82 | 91.60±1.06 | 91.77±1.11 | /// | /// |
| TS-TCC | 97.23±0.10 | 95.54±0.08 | 90.37±0.34 | 90.38±0.39 | 83.00±0.71 | 73.57±0.74 |
| TSTCC-Floss | 97.41±0.17 | 97.75±0.00 | 90.86±0.34 | 90.56±0.35 | 83.70±0.45 | 73.53±0.39 |
| TF-C | 96.21±0.25 | 95.23±0.21 | 88.15±1.32 | 88.75±0.52 | 78.82±0.64 | 72.48±0.56 |
| BTSF | 95.46±0.35 | 94.85±0.16 | 85.44±0.26 | 85.51±1.13 | 73.94±0.23 | 71.62±0.74 |
| CA-TCC | 97.74±0.38 | 97.01±0.13 | 91.68±0.46 | 91.79±0.24 | 84.04±0.42 | 74.42±0.68 |
| **TFCC(ours)** | **98.15±0.17** | **97.92±0.35** | **92.73±0.57** | **92.52±0.59** | **84.35±0.52** | **75.60±0.45** |

Forecasting: We conduct a comprehensive linear evaluation to access the performance on four benchmarks across a range of forecasting lengths $T \in \{24, 48, 168, 336, 720\}$ using two metrics: the mean square error (MSE) and the mean absolute error (MAE). Specifically, we designed a linear regression model with an L2 paradigm penalty, trained on top of a fine-tuned self-supervised pre-trained encoder. The training phase includes two stages: (1) learning representations through the TFCC framework, and (2) training a linear regressor for each $T$ on top of the learned representations. Table 2 shows that TFCC excels in multivariate forecasting, outperforming all baselines across varying datasets and lengths. Notably, TFCC's performance improves with increasing sequence length compared to the baselines, indicating our model can better use global context information and is more capable of capturing long-term dependencies in time series than other baseline models. More forecasting results compared to other baselines are in Supplementary Materials.

### 4.3 Ablation study

To assess the effectiveness of each component in TFCC, we conducted ablation studies by comparing TFCC and its different variants. Table 3 shows the ablation study on four datasets. We can see the temporal cross-view forecasting task boosts feature robustness, leading to ~2% improvement in Epilepsy and ~6% in HAR, ~10% in ETTh1 and 3% in Weather (in Accuracy and MSE). Additionally, introducing contextual consistency (temporal, frequency) further enhances performance by strengthening feature discriminability with ~1% boost in Epilepsy and 16.6% in ETTh1. Performance improvements are observed when adding frequency cross-view forecasting, with further 0.59% and 1.11% improvement in Epilepsy and HAR, 30.9% and 20.6% in ETTh1 and Weather, meaning frequency can help to learn periodic features of time series. Furthermore, incorporating temporal-frequency consistency in our TFCC framework demonstrates the best performance, further enhancing the model's ability to capture discriminative representations. Last, our TFCC with both T-Aug and F-Aug is superior to only using T-Aug or F-Aug. These findings suggest that each self-supervised contrasting view contributes significantly to learning more useful and valuable representations.

### 4.4 Transfer learning

We further validate the transferability of our proposed method by designing transfer learning experiments to verify whether it learns general representations that can be applied to different domains. We evaluate the transferability using the Fault Diagnosis (FD) datasets from four different domains, denoted as domains A, B, C, and D. we select one domain (source domain) for training and test it on the other three domains (target domains). Our experiment performs two steps on the source domain: (1) self-supervised training to get a pre-trained encoder, and (2) fine-tuning on the pre-trained encoder. Finally, we utilize the fine-tuned model to obtain accuracy on the test set in each target domain. Table 4 shows the performance of supervised, TS-TCC, and TFCC in 12 cross-domain scenarios. The representations of our fine-tuned method on the source domain can be well adapted to the target domains. TFCC achieves best in 9 out of 12 cross-domain scenarios. Notably, TFCC has 100% accuracy in B → D and D → B. Overall, TFCC improves the transferability of the learned representations over the supervised, TS-TCC, and CA-TCC training, by ~8%, ~4%, and ~2% in terms of accuracy in 12 scenarios, respectively.

### 4.5 Semi-supervised Analysis

To evaluate the effectiveness of TFCC in a semi-supervised setting, we fine-tuned the pre-trained encoder with 1%, 5%, 10%, 50%, 75%, and 100% of randomly selected instances of training data. Figure 4 presents the results of our fine-tuned TFCC (i.e., blue curves) compared to supervised training on the three classification datasets under the above settings. The results indicate our model performs significantly better than supervised training with limited labeled data. For example, with only 1% labeled data, TFCC fine-tuning still achieves 76.04% in HAR and 90.74% in Epilepsy. Especially for the unbalanced Sleep-EDF, our method can also reach 65.25%, while supervised training is only 32.24%. Moreover, TFCC's performance steadily improves as the percentage of labeled data increases, proving its applicability to label-scarce environments.

### 4.6 Sensitivity Analysis

TFCC incorporates several hyperparameters requiring careful tuning. To achieve optimal performance, we conducted a sensitivity

**Table 2: Multivariate forecasting results. Best results are highlighted in bold, while the second-best is underlined.**

| Methods | | Representation Learning | | | | | | | | | | | End-to-end Forecasting | | | |
|---|---|---|---|---|---|---|---|---|---|---|---|---|---|---|---|---|
| | | TFCC | | TNC | | TS-TCC | | CoST | | TS2Vec | | BTSF | | TCN | | Informer | |
| Metric | | MSE | MAE | MSE | MAE | MSE | MAE | MSE | MAE | MSE | MAE | MSE | MAE | MSE | MAE | MSE | MAE |
| ETTh1 | 24 | **0.271** | **0.428** | 0.708 | 0.592 | 0.653 | 0.610 | 0.386 | 0.429 | 0.590 | 0.534 | 0.541 | 0.519 | 0.583 | 0.547 | 0.577 | 0.549 |
| | 48 | **0.417** | **0.452** | 0.749 | 0.619 | 0.720 | 0.693 | 0.437 | 0.464 | 0.624 | 0.555 | 0.613 | 0.524 | 0.670 | 0.606 | 0.685 | 0.625 |
| | 168 | **0.575** | **0.524** | 0.884 | 0.699 | 1.129 | 1.044 | 0.643 | 0.582 | 0.762 | 0.639 | 0.640 | 0.532 | 0.811 | 0.680 | 0.931 | 0.752 |
| | 336 | **0.789** | **0.677** | 1.020 | 0.768 | 1.492 | 1.076 | 0.812 | 0.679 | 0.931 | 0.728 | 0.864 | 0.689 | 1.132 | 0.815 | 1.128 | 0.873 |
| | 720 | **0.923** | **0.711** | 1.157 | 0.830 | 1.603 | 1.206 | 0.970 | 0.771 | 1.063 | 0.790 | 0.993 | 0.712 | 1.165 | 0.813 | 1.215 | 0.896 |
| ETTh2 | 24 | **0.312** | **0.458** | 0.612 | 0.595 | 0.883 | 0.747 | 0.447 | 0.502 | 0.423 | 0.489 | 0.663 | 0.557 | 0.935 | 0.754 | 0.72 | 0.665 |
| | 48 | **0.618** | 0.636 | 0.840 | 0.716 | 1.701 | 1.378 | 0.699 | 0.637 | 0.619 | **0.605** | 1.245 | 0.897 | 1.300 | 0.911 | 1.457 | 1.001 |
| | 168 | **1.161** | **0.838** | 2.359 | 1.213 | 3.956 | 2.301 | 1.549 | 0.982 | 1.845 | 1.074 | 2.669 | 1.393 | 4.017 | 1.579 | 3.489 | 1.515 |
| | 336 | **1.218** | **0.862** | 2.782 | 1.349 | 3.992 | 2.852 | 1.749 | 1.042 | 2.194 | 1.215 | 1.954 | 1.093 | 3.460 | 1.456 | 2.723 | 1.340 |
| | 720 | **1.209** | **0.938** | 2.753 | 1.394 | 4.732 | 2.345 | 1.971 | 1.092 | 2.636 | 1.373 | 2.566 | 1.276 | 3.106 | 1.381 | 3.467 | 1.473 |
| ETTm1 | 24 | 0.312 | 0.359 | 0.522 | 0.472 | 0.473 | 0.490 | **0.246** | **0.329** | 0.453 | 0.436 | 0.302 | 0.342 | 0.363 | 0.397 | 0.323 | 0.369 |
| | 48 | 0.331 | 0.382 | 0.695 | 0.567 | 0.671 | 0.665 | **0.331** | 0.386 | 0.592 | 0.515 | 0.395 | 0.387 | 0.542 | 0.508 | 0.494 | 0.503 |
| | 168 | **0.370** | **0.387** | 0.731 | 0.595 | 0.803 | 0.724 | 0.378 | 0.419 | 0.635 | 0.549 | 0.438 | 0.399 | 0.666 | 0.578 | 0.678 | 0.614 |
| | 336 | **0.411** | 0.514 | 0.818 | 0.649 | 1.958 | 1.429 | 0.472 | 0.486 | 0.693 | 0.609 | 0.675 | **0.429** | 0.991 | 0.735 | 1.056 | 0.786 |
| | 720 | **0.463** | **0.552** | 0.932 | 0.712 | 1.838 | 1.601 | 0.620 | 0.574 | 0.782 | 0.655 | 0.721 | 0.643 | 1.032 | 0.756 | 1.192 | 0.926 |
| Weather | 24 | **0.296** | **0.357** | 0.320 | 0.373 | 0.572 | 0.603 | 0.298 | 0.360 | 0.307 | 0.363 | 0.324 | 0.369 | 0.321 | 0.367 | 0.335 | 0.381 |
| | 48 | 0.405 | 0.462 | 0.380 | 0.421 | 0.647 | 0.691 | **0.359** | **0.411** | 0.374 | 0.418 | 0.366 | 0.427 | 0.386 | 0.423 | 0.395 | 0.459 |
| | 168 | **0.458** | **0.466** | 0.479 | 0.495 | 1.117 | 0.962 | 0.464 | 0.491 | 0.491 | 0.506 | 0.543 | 0.477 | 0.491 | 0.501 | 0.608 | 0.567 |
| | 336 | **0.462** | **0.467** | 0.505 | 0.514 | 1.783 | 1.370 | 0.497 | 0.517 | 0.525 | 0.530 | 0.568 | 0.487 | 0.502 | 0.507 | 0.702 | 0.620 |
| | 720 | 0.644 | **0.493** | 0.543 | 0.547 | 1.850 | 1.566 | 0.533 | 0.542 | 0.556 | 0.552 | 0.601 | 0.522 | **0.498** | 0.508 | 0.831 | 0.731 |
| Avg. | | **0.582** | **0.548** | 0.989 | 0.706 | 1.629 | 1.218 | 0.693 | 0.585 | 0.855 | 0.657 | 0.884 | 0.634 | 1.149 | 0.741 | 1.150 | 0.782 |

**Table 3: Ablation study of each component in TFCC. (1) "T only" denotes the temporal contrasting module without cross-view forecasting, (2) "T+T-Aug" signifies adding temporal cross-view forecasting, (3) "T+TC+T-Aug" adds temporal consistency, (4) "T+TC+FC+T-Aug" adds frequency consistency, (5) "T+TC+FC+X-Aug" adds frequency cross-view forecasting, (6) "TFCC(T+TC+FC+TFC+X-Aug)" is our framework, and (7) "TFCC(T-Aug only)" and (8) "TFCC(F-Aug only)" are TFCC with only time augmentations and frequency augmentations without cross-view forecasting.**

| | Epilepsy | | HAR | | ETTh1 | | Weather | |
|---|---|---|---|---|---|---|---|---|
| Component | Accuracy | MF1 | Accuracy | MF1 | MSE | MAE | MSE | MAE |
| T only | 94.39±1.19 | 90.93±1.41 | 82.76±1.50 | 82.17±1.64 | 0.594 | 0.559 | 0.435 | 0.562 |
| T+T-Aug | 96.69±0.69 | 94.61±0.76 | 88.46±2.30 | 88.40±2.26 | 0.536 | 0.517 | 0.422 | 0.517 |
| T+TC+T-Aug | 97.43±0.61 | 95.86±0.92 | 90.61±0.64 | 90.63±0.98 | 0.529 | 0.535 | 0.414 | 0.461 |
| T+TC+FC+T-Aug | 97.65±0.26 | 96.12±0.59 | 91.10±1.82 | 91.04±1.85 | 0.447 | 0.528 | 0.412 | 0.456 |
| T+TC+FC+X-Aug | 98.04±0.32 | 97.23±0.47 | 91.72±1.61 | 91.60±1.15 | 0.309 | 0.459 | 0.327 | 0.442 |
| TFCC (T+TC+FC+TFC+X-Aug) | **98.15±0.17** | **97.92±0.35** | **92.73±0.57** | **92.52±0.59** | **0.271** | **0.428** | **0.296** | **0.357** |
| TFCC (T-Aug only) | 97.55±0.49 | 95.49±0.32 | 89.60±2.26 | 88.43±2.23 | 0.324 | 0.471 | 0.391 | 0.452 |
| TFCC (F-Aug only) | 97.30±0.85 | 95.32±0.41 | 88.48±1.38 | 86.21±1.44 | 0.331 | 0.466 | 0.411 | 0.460 |

**Table 4: Cross-Domain transfer learning results on Fault Diagnosis dataset**

| Method | FD-A | | | FD-B | | | FD-C | | | FD-D | | | AVG |
|---|---|---|---|---|---|---|---|---|---|---|---|---|---|
| | A→B | A→C | A→D | B→A | B→C | B→D | C→A | C→B | C→D | D→A | D→B | D→C | |
| Supervised | 34.48 | 44.94 | 34.57 | **52.93** | 63.67 | 99.82 | **52.93** | 84.02 | 83.54 | 53.15 | 99.56 | 62.43 | 63.83 |
| TS-TCC | 43.15 | 51.50 | 42.74 | 47.98 | 70.38 | 99.30 | 38.89 | 98.31 | 99.38 | 51.91 | 99.96 | 70.31 | 67.82 |
| CA-TCC | 44.75 | 52.09 | 45.63 | 46.26 | 71.33 | **100.0** | 52.71 | **99.85** | **99.84** | 46.48 | **100.0** | 77.01 | 69.66 |
| TFCC | **55.91** | **53.85** | **62.72** | 48.43 | **71.45** | **100.0** | 46.61 | 98.84 | 97.98 | **53.68** | **100.0** | 71.11 | **71.72** |

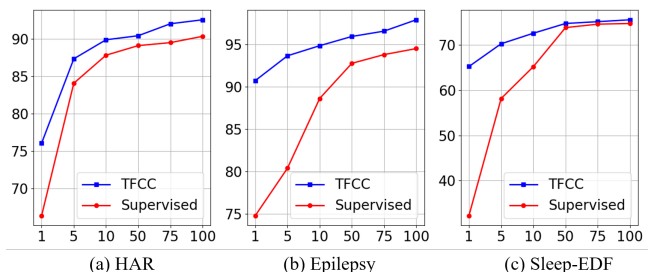

**Figure 4: Comparison between TFCC fine-tuning vs. supervised for different percentages of few labels in MF1-score.**

analysis on two datasets, HAR and ETTh1, focusing on five key parameters: dropout mask ratio, temperature $\tau$, and the weights of contrastive losses $\lambda1$, $\lambda2$, and $\lambda3$. As delineated in Table 5, optimal performance is attained with a dropout mask ratio of 0.2. This is because too high value may lose the original properties of time series and too low may lead to representation collapse. Table 6 shows that TFCC gets the best performance when $\tau$= 0.15. A reasonable $\tau$ will facilitate the optimization of the training process and make the representation more discriminative as it is adjusted. Figure 5 explores the influence of $\lambda1$, $\lambda2$, and $\lambda3$ on ETTh1 under length=24. We first fixed $\lambda1$ at 0.1 and varied $\lambda2$ and $\lambda3$. The results suggest that the model performs best at $\lambda2$ = 0.6 and $\lambda3$ = 0.4. Subsequently, we fixed the two values and tuned the value of $\lambda1$. Once again, a moderate value of $\lambda1$ = 0.1 yielded the best outcomes.

**Table 5: Sensitivity experiments of drop_mask ratio**

| drop_mask | p=0.01 | p=0.1 | p=0.15 | p=0.2 | p=0.3 | p=0.5 |
|---|---|---|---|---|---|---|
| HAR | 87.73 | 90.58 | 91.39 | **92.73** | 91.21 | 89.10 |
| ETTh1(24, MSE) | 0.287 | 0.282 | 0.280 | **0.271** | 0.284 | 0.311 |

**Table 6: Sensitivity experiments of $\tau$**

| $\tau$ | 0.01 | 0.1 | 0.15 | 1 | 10 | 100 |
|---|---|---|---|---|---|---|
| HAR | 90.07 | 91.88 | **92.73** | 91.72 | 90.94 | 90.63 |
| ETTh1(24, MSE) | 0.319 | 0.301 | **0.271** | 0.32 | 0.314 | 0.308 |

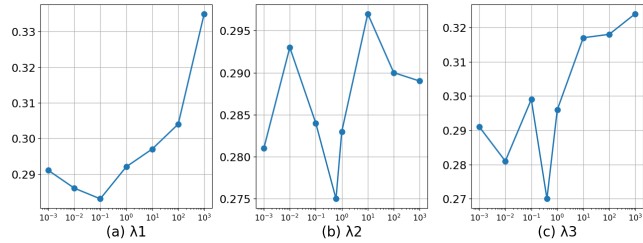

**Figure 5: Sensitivity analysis of $\lambda1$, $\lambda2$ and $\lambda3$ on ETTh1.**

## 4.7 Augmentation Analysis

Augmentations selection is less explored, which is still an open and pending problem. Table 7 presents the augmentation effects. it can be seen that MSE and MAE reached 0.346 and 0.482 when slicing was used alone. Furthermore, the MSE increased from 0.297 to 0.301, MAE rose from 0.453 to 0.464 when slicing is used in multi-augmentations, with worse performance, which may be due to slicing destroying the long-term dependence, leading to sampling bias. In contrast, when adding "drop_mask" and "phase perturbation", the model's performance has a significant improvement, with the MSE decreasing from 0.304 to 0.294, and further to 0.281. Specifically, when adding phase perturbation, the MSE decreased from 0.294 to 0.271, and the MAE decreased from 0.451 to 0.428, which are improved by 7.8% and 5.1%, respectively, indicating that the introduction of phase perturbation can significantly improve the performance of the model. These findings reveal that TFCC can produce representations invariant to temporal and spectrum perturbations, and appropriate time and frequency augmentations can improve model generalization.

**Table 7: Augmentation Analysis on ETTh1 (24)**

| Time Augmentation | Frequency Augmentation | MSE | MAE |
|---|---|---|---|
| slicing | add | 0.346 | 0.482 |
| scale | add | 0.321 | 0.463 |
| scale, jitter | add, remove | 0.303 | 0.453 |
| drop_mask | add, remove | 0.304 | 0.459 |
| scale, jitter, drop_mask | add,remove | 0.294 | 0.451 |
| slicing, jitter, drop_mask | remove, phase_pertub | 0.301 | 0.464 |
| jitter, drop_mask | add, remove,phase_pertub | 0.281 | 0.439 |
| jitter, drop_mask | remove, phase_pertub | 0.297 | 0.453 |
| scale, jitter, drop_mask | add, remove, phase_pertub | **0.271** | **0.428** |

## 5 CONCLUSION

This study proposes TFCC, a novel framework specifically engineered for unsupervised representation learning from time series data. In TFCC, we use the entire series as input and construct two sets of instance-level augmentation families based on the temporal and frequency characteristics to generate different views for training. Then, the framework injects multi-view contrasting modules into pre-training to learn properties invariant to diverse perturbations within various complex dynamics. Specifically, two challenging cross-view forecasting tasks between the original and its augmented series in time and frequency domains are first designed for robust representation learning. Contextual consistency brings the temporal-based and frequency-based representations along with their local neighbors close together in their latent space. Temporal-frequency consistency fosters mutual representation learning automatically and unveils temporal-spectral correlations through cross-domain dependencies, thus promoting discriminative representation learning. Comprehensive experiments validate TFCC's exceptional prowess in both classification and forecasting tasks, alongside its notable efficiency in label-scarce and transfer learning scenarios.

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
