# OpenReview forum: "Multi-view Self-Supervised Contrastive Learning for Multivariate Time Series"
_acmmm.org/ACMMM/2024/Conference — MM2024 Poster_

### Official Review · Reviewer_dbrp · 2024-05-20

**Rating:** 3
**Confidence:** 3

**Summary:**

This paper proposes Multi-view Self-Supervised Contrastive Learning for Multivariate Time Series

**Strengths:**

1. well written
2. Ample experiment

**Limitations:**

1. batch size is a very important hyperparameter for Contrastive Learning. I suggest the author evaluate it

2. Lack of comparison with state-of-the-art baselines [1] [2] [3].

3. I believe that the transferability of representation learning is crucial. Can TFCC achieve good performance with other backbone models as well?

4. The idea of integrating both time-domain and frequency-domain concepts has been previously explored in related works [4,5], and the authors are not the first to propose it.

[1] SimMTM: A Simple Pre-Training Framework for Masked Time-Series Modeling

[2] A TIME SERIES IS WORTH 64 WORDS: LONG-TERM FORECASTING WITH TRANSFORMER

[3] Parametric Augmentation for Time Series Contrastive Learning

[4] Self-supervised contrastive pre-training for time series via time-frequency consistency

[5] TimesURL: Self-Supervised Contrastive Learning for Universal Time Series Representation Learning

**Suitability:**

2

---

### Official Review · Reviewer_eosU · 2024-05-24

**Rating:** 4
**Confidence:** 3

**Summary:**

This paper propose a contrastive learning-based method to learn the representation of time series data by leveraging both time and frequency domain information.

**Strengths:**

1. The paper is well written and easy to follow.

2. Compared with the frequency domain-based contrastive learning, the paper considers the phase information.

3. Extensive experiments demonstrate that the method achieves superior performance in classification and forecasting.

**Limitations:**

1.  The important claims seem to be lack of visualizations and deep analysis. For example , why the phase information is important for the representation learning? And what a role it played in the representation learning?

2. The baselines chosen for time series forecasting are too old. Informer is published in 2021 and TCN is in 2018.

**Suitability:**

2

---

### Official Review · Reviewer_UVhz · 2024-05-24

**Rating:** 3
**Confidence:** 3

**Summary:**

This paper addresses issues of sampling bias and suboptimal performance caused by the loss of global context and neglect of crucial frequency information by introducing a self-supervised framework named Temporal-Frequency and Contextual Consistency (TFCC). The framework incorporates dual instance-level augmentations across the entire series, dual cross-view forecasting tasks in both time and frequency dimensions, and three specially designed consistency modules—temporal, frequency, and temporal-frequency—to robustly enhance data representation.

**Strengths:**

(1) This paper introduces a method named TFCC that employs instance-level augmentations in both time and frequency domains, specifically through frequency spectrum perturbation. This approach retains the raw properties of the series and effectively reduces false negatives and false positives, thereby enhancing the accuracy of the contrastive learning model.

(2) The paper designs two cross-view forecasting tasks aimed at developing robust representations, and strengthening the framework's analytical capabilities.

(3) The paper provides a comprehensive set of experiments to support its findings.

**Limitations:**

(1) The paper lacks innovation, as the fusion method combining time and frequency domains has been previously addressed, notably in the study "Unsupervised Time-Series Representation Learning with Iterative Bilinear Temporal-Spectral Fusion."

(2) The enhancement techniques employed in this paper, specifically the temporal contrast learning method, have already been explored in other research, such as TS2Vec: Towards Universal Representation of Time Series.

(3) The focus of the paper on single modality does not align well with the main theme of the conference.

**Suitability:**

3

---

### Official Review · Reviewer_gSU9 · 2024-05-25

**Rating:** 4
**Confidence:** 3

**Summary:**

This paper introduces TFCC, a multi-view self-supervised contrastive learning framework for time series representation learning. TFCC addresses limitations of existing methods by incorporating both temporal and frequency domain information. It leverages instance-level augmentations, cross-view forecasting in both domains, and consistency modules to learn robust and discriminative representations. Experiments demonstrate TFCC's slightly improved performance compared to state-of-the-art methods in classification, forecasting, semi-supervised learning, and transfer learning settings.

**Strengths:**

1) The paper leverages both temporal and frequency domain information for time series representation learning. This approach tackles limitations of existing methods that focus solely on the temporal domain or fail to fully exploit frequency information.

TFCC combines various elements:
- Instance-level augmentations for preserving global context and long-term dependencies.
- Cross-view forecasting in both time and frequency domains to learn robust representations.
- Contextual consistency modules for discriminative learning.
- Temporal-frequency consistency for aligning time-based and frequency-based embeddings.

2) The paper presents extensive experiments on diverse datasets covering both classification and forecasting tasks. It also demonstrates effectiveness in semi-supervised and transfer learning settings.
3) The paper provides a thorough explanation of the proposed framework, including the motivations behind each component and its contribution to the overall performance.
4) The authors perform thorough ablation studies to evaluate the impact of each component of the framework, providing valuable insights into the effectiveness of individual modules.

**Limitations:**

1) While the paper mentions sensitivity analysis, it lacks a detailed discussion on the process of hyperparameter tuning and how it impacts performance. It appears that if the hyperparameters are not optimized, the performance of the model can fall below the state of the art.
2) The paper could benefit from a discussion on potential overfitting issues related to the complex architecture and multiple loss functions.
3) The paper does not address the computational complexity and scalability of the TFCC framework, particularly for large-scale datasets or high-dimensional time series. The proposed framework adds considerable complexity compared to vanilla contrastive learning methods, for only modest improvements in accuracy. It is unclear if the accuracy improvements justify the added complexity.
4) The paper acknowledges augmentation selection as an open problem, but provides limited analysis on the impact of different augmentation strategies on model performance.
5) The paper primarily focuses on benchmark datasets. Discussing potential applications of TFCC in real-world scenarios and providing case studies would enhance the paper's impact.

**Suitability:**

2

---

### Meta-Review · Area_Chair_Fm3S · 2024-07-09

**Recommendation:** Accept (Poster)
**Confidence:** 3

**Metareview:**

This paper introduces TFCC, a multi-view self-supervised contrastive learning framework for time series representation learning, incorporating both temporal and frequency domain information. While reviewers appreciated the comprehensive experiments, clear writing, and innovative combination of temporal and frequency domain augmentations, they raised several concerns. Key limitations include the modest novelty of the approach, lack of detailed hyperparameter tuning, potential overfitting issues, and limited discussion on computational complexity and scalability. Additionally, the baselines used for comparison are outdated, and the paper lacks visualization and deep analysis of key claims. Despite these issues, the paper demonstrates slight improvements over state-of-the-art methods and has potential applications in real-world scenarios. The ratings are mixed.

After carefully reading the comments and the paper, AC leans towards borderline acceptance. Some reviewers challenged the work to be not  well aligned with MM theme, which AC finds it aligned.